# Brain adiponectin signaling controls peripheral insulin response in *Drosophila*

Nathalie Arquier [1✉], Marianne Bjordal[2], Philippe Hammann [3], Lauriane Kuhn [3] & Pierre Léopold[4]

The brain plays a key role in energy homeostasis, detecting nutrients, metabolites and circulating hormones from peripheral organs and integrating this information to control food intake and energy expenditure. Here, we show that a group of neurons in the *Drosophila* larval brain expresses the adiponectin receptor (AdipoR) and controls systemic growth and metabolism through insulin signaling. We identify glucose-regulated protein 78 (Grp78) as a circulating antagonist of AdipoR function produced by fat cells in response to dietary sugar. We further show that central AdipoR signaling inhibits peripheral Juvenile Hormone (JH) response, promoting insulin signaling. In conclusion, we identify a neuroendocrine axis whereby AdipoR-positive neurons control systemic insulin response.

[1] Aix-Marseille University, INSERM, TAGC, UMR_S1090, Marseille, France. [2] Université Côte d'Azur, CNRS UMR7277, Inserm U1091, iBV, Parc Valrose, 06108 Nice, France. [3] Plateforme Protéomique Strasbourg-Esplanade FRC 1589, University of Strasbourg, CNRS, 15 rue René Descartes, 67084 Strasbourg, cedex, France. [4] Institut Curie, PSL Research University, CNRS UMR3215, INSERM U934, 26 Rue d'Ulm, 75005 Paris, France. ✉email: nathalie.arquier@univ-amu.fr

Energy homeostasis is controlled by a complex interplay between peripheral organs and the brain. In the vertebrate brain, nuclei within the hypothalamus crosstalk and integrate peripheral signals such as the level of adiposity and caloric intake to regulate food intake and energy expenditure[1]. Glucose being the primary source of energy, its availability is constantly monitored through sensor and effector mechanisms[2]. Adiponectin is an abundant adipokine present in the bloodstream of mice and humans, whose levels inversely correlate with circulating glucose[3]. Adiponectin receptor expression also inversely correlates with insulinemic and glycemic states in mice[4], suggesting that adiponectin signaling could serve as a marker for insulin sensitivity. In line with this, *Adiponectin* knockout mice fed a high-fat/high-sucrose diet develop severe insulin resistance[5]. Adiponectin can cross the blood–brain barrier[6,7], and one isoform of its receptor, AdipoR1, is expressed in the hypothalamus in humans[8]. However, conflicting reports have confounded the evaluation of the role of brain adiponectin signaling in energy homeostasis. Our current study using Drosophila identifies crosstalk between the brain and peripheral organs whereby adiponectin receptor-positive neurons receive nutritional inputs and control general insulin signaling by modulating the levels of peripheral juvenile hormone (JH) response.

## Results and discussion

**Brain adiponectin receptor controls animal growth.** *Drosophila* has a single adiponectin receptor, AdipoR, encoded by gene *CG5315* (referred to as *AdipoR*). Independent batches of AdipoR antibodies labeled the same symmetrical cluster of neurons in the larval brain lobes only (Fig. 1a and Supplementary Fig. 1a–e), referred to as adiponectin receptor-positive neurons (APNs). Unlike a previous report[9], our anti-AdipoR antibodies did not label the brain insulin-producing cells (IPCs) (Supplementary Fig. 1b). The pattern of Gal4 expression of Janelia GAL4 line #48522 (see Methods), was found to overlap to a large extent with the APNs (Supplementary Fig. 1a) and was further referred to as *Apn-GAL4* (*Apn >*). RNAi-mediated silencing of *AdipoR* in APNs using two independent RNAi lines (*Apn > AdipoR-Ri* and *Apn > AdipoR-TRIP*) induced partial larval lethality with a reduced imaginal disc (−48%, Fig. 1b) and larval body growth (Supplementary Fig. 1f). Emerging adults had smaller wings (−10%, Fig. 1c and Supplementary Fig. 1g) and reduced weight (−20%, Fig. 1d and Supplementary Fig. 1h), accompanied with a 63% reduction of anti-AdipoR staining in the APNs (Supplementary Fig. 1c–c′). Reducing *AdipoR* in the IPCs had no effect on growth (*dilp2 > AdipoR-Ri*; Supplementary Fig. 1i) suggesting an effect on growth control independent of IPC function. Activating *AdipoR* in the APNs induced growth increase (+12%, Fig. 1d, *Apn > AdipoR-act*, see Supplementary materials for details). Similarly, expression of human adiponectin (hAdipoQ), the vertebrate AdipoR agonist, from larval fat cells or APNs, induced systemic growth with adults heavier than controls (*lpp > hAdipoQ*, +7%; Fig. 1d; *Apn > hAdipoQ*, +14%, Supplementary Fig. 1k). Moreover, a strong hypomorphic mutant for *AdipoR* (*AdipoR^M*, Supplementary Fig. 1n) exhibited reduced larval growth (Supplementary Fig. 1m) and developmental lethality in accordance with the *Apn > AdipoR-Ri* phenotypes. Reducing *AdipoR* function in the larval fat body or in the muscles, two organs with adiponectin signaling function in vertebrates[10], induced a moderate increase in adult size, which did not account for the *AdipoR^M* phenotype (Supplementary Fig. 1l). Collectively, these results indicate that *Drosophila* AdipoR acts in the brain to control systemic growth.

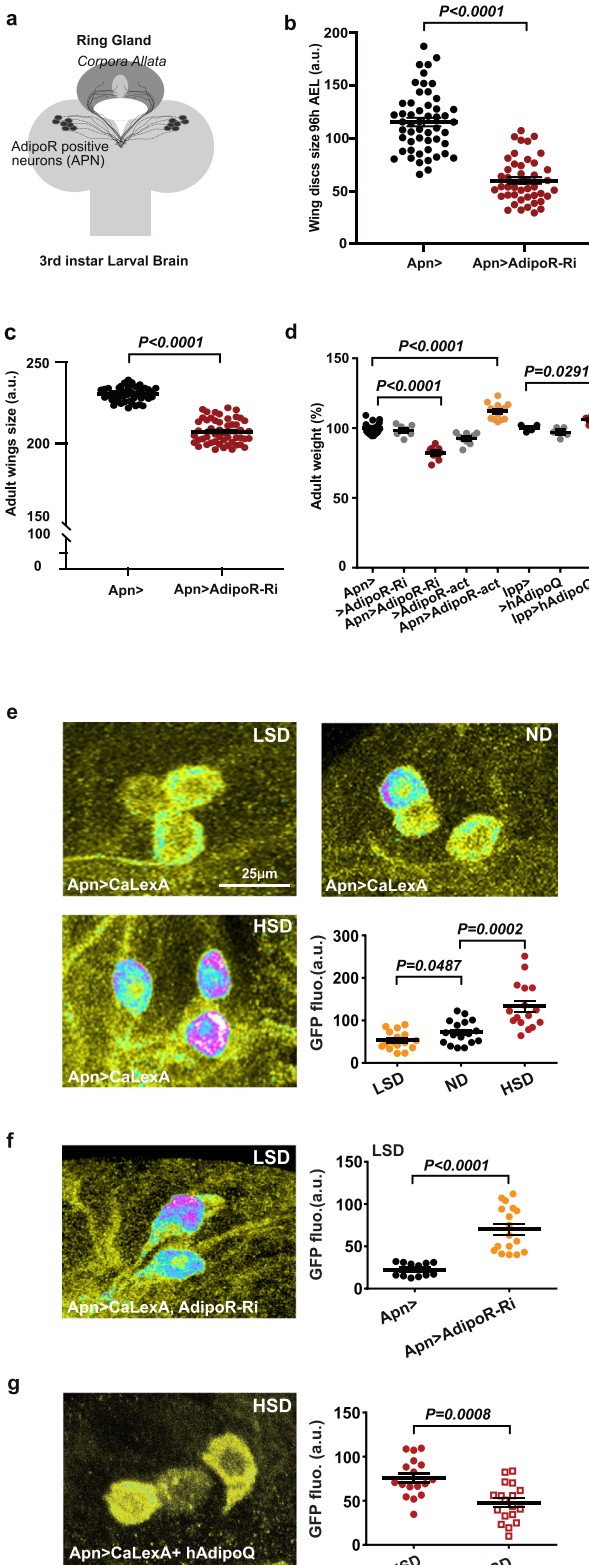

**AdipoR controls APN activity according to dietary sugars.** The role of vertebrate AdipoR signaling in energy homeostasis suggested that the fly APNs could function in sensing food caloric content. Using the calcium reporter *UAS-CaLexA* as a marker for synaptic activity[11], we observed that activation of APNs gradually increased with dietary sugar content (Fig. 1e). In addition, silencing *AdipoR* in APNs increased neuronal activity in larvae

**Fig. 1 Central AdipoR function controls animal growth. a** Scheme of AdipoR-positive neurons (APNs) and their projections on the corpora allata of the ring gland. **b** Measure of larval wing disc size 96 h after egg laying (AEL) from *Apn > AdipoR-Ri* (*n* = 45, red) and *Apn >* (*n* = 52, black). **c** Measure of adult wing size of *Apn > AdipoR-Ri* (*n* = 51, red) and *Apn >* (*n* = 47, black). *n* = biologically independent samples from five (**b**) and three (**c**) independent experiments. Statistical significance was tested using ordinary two-way ANOVA with Sidak's multiple comparisons test. **d** Measure of adult weight of *Apn > AdipoR-Ri* (*n* = 9, red), *Apn >* (*n* = 20, black), *>AdipoR-Ri* (*n* = 9, gray), *>AdipoR-act* (*n* = 10, gray), *Apn > AdipoR-act* (*n* = 14, orange), *lpp >* (*n* = 4, black), *>hAdipoQ* (*n* = 5, gray) and *lpp >hAdipoQ* (*n* = 5, red). *n* = groups of 5–10 males. Statistical significance was tested ordinary one-way ANOVA with Tukey's multiple comparisons test. **e** APN activity levels at 96 h AEL revealed by anti-GFP immunostaining and quantified by GFP fluorescence intensity using a *UAS-CaLexA* probe (see Methods) in different sugar diets (LSD, ND, HSD), upon modulation of AdipoR expression in LSD (in **f**) or upon human AdipoQ feeding (in **g**). Quantifications are shown in accompanying panels (**e**, LSD *n* = 16, orange; ND *n* = 17, black; HSD *n* = 16, red), (**f**, *Apn > AdipoR-Ri n* = 17, orange; *Apn > n* = 12, black), (**g**; +hAdipoQ *n* = 20, red squares; control *n* = 18, red points). *n* = number of independent GFP measurements. Statistical significance was tested using a two-sided Mann–Whitney test. Data were presented as mean values ± SEM. *P* values are indicated in all panels. a.u. arbitrary units. Source data are provided as a Source Data file.

fed a low-sugar diet (LSD) (Fig. 1f), whereas feeding larvae on a high-sugar diet (HSD) supplemented with human adiponectin (+hAdipoQ) prevented APNs activation (Fig. 1g). We concluded that AdipoR function blocks APNs activity and that human adiponectin acts as an agonist of the fly AdipoR (Fig. 1d and Supplementary Fig. 1k) to prevent APNs activation by dietary sugar. Therefore, an endogenous agonist of AdipoR could be produced in LSD, maintaining APNs silent, or AdipoR could be repressed by an antagonist partner in response to HSD leading to APN activation, or both regulations could take place at various dietary sugar levels.

**Grp78 is an antagonist partner of AdipoR.** The fly genome does not encode a peptide with adiponectin-related sequences. Therefore, we conducted IP-mass spectrometry (IP-MS) on larvae ubiquitously expressing Myc-tagged AdipoR from a tubulin promoter (*tub > AdipoR-Myc*) to identify endogenous interactors of AdipoR. Following anti-Myc immunoprecipitation, larval lysate was subjected to MS analysis and AdipoR-associated proteins were identified (Supplementary Fig. 2a, b and Supplementary Materials). IP-MS specificity was tested by mock IP with larval extracts expressing an unrelated seven-pass membrane-associated protein (Smo-myc; see Methods). Genes encoding secreted or putative secreted products were validated by knockdown in fat cells, and pupal lethality was scored (Supplementary Fig. 2c and Supplementary Materials). Among the putative candidates, glucose-regulated protein 78 (Grp78, also named BIP or HSC70-3), is regulated by sugar both in mammals and flies[12,13]. Antibodies raised against the mammalian homolog[12] detected an increase of Grp78 amount in the hemolymph of larvae raised on HSD compared to LSD (Fig. 2a and Supplementary Fig. 2d, e). Grp78 is a component of the unfolded protein response (UPR)[14]. However, examining *Xbp1* splicing and its nuclear localization in fat body cells as markers of UPR, we found that UPR is not activated in our HSD conditions (Supplementary Fig. 2f–h), suggesting that Grp78 acts on adiponectin signaling independently of UPR. *grp78* mRNA levels were not modified by our dietary sugar conditions (Fig. 2b), in line with a regulation of Grp78 function independent of UPR and *ATF6* transcriptional

activation[15]. Reciprocal co-immunoprecipitation experiments in S2 cells also suggested specific interactions between AdipoR, hAdipoQ, and Grp78 (Supplementary Fig. 1j, Fig. 2c). Finally, when dissected brains from larvae fed on LSD were cultured in the presence of hemolymph from larvae fed on HSD, increased APNs activity was observed (Fig. 2d–d′), providing an ex-vivo confirmation of in vivo results (Fig. 1e). However, increased APNs activity was not observed when hemolymph was collected from larvae fed on HSD with fat body-specific *grp78* silencing (*lpp > grp78-Ri*, Fig. 2d–d′). This indicates that Grp78 is an antagonistic partner of AdipoR produced by the adipose tissue and required for APNs activation in high sugar conditions.

**Antagonistic interaction between AdipoR and JH signaling.** APNs send numerous projections towards a key endocrine organ called the ring gland (RG) (Fig. 3a), which produces three metabolic hormones, adipokinetic hormone (AKH), JH, and ecdysone. We then examined the role of these hormones as a potential peripheral effector for the central function of AdipoR. Silencing *AdipoR* in APNs (*Apn > AdipoR-Ri*) resulted in increased expression of JH-inducible protein 26 (*jhi-26*), a target of JH signaling[16] (Fig. 3b). A similar result was observed for Kr-h1, the transcriptional effector for JH signaling (Fig. 3c). This suggested that inhibiting *AdipoR* in the APNs leads to elevated JH signaling. In this line, as observed for *Apn > AdipoR-Ri* animals, *wt* larvae fed with *pyriproxyfen*, a potent JH analog (JHa)[17], grew slowly (−69% in imaginal wing disc size, Fig. 3d) and adult escapers exhibited weight reduction (−6%, Fig. 3e). Hemizygous females mutant for both *met* and *gce* (encoding the two JH receptors in *Drosophila*[18]), showed survival rates similar to controls (+/FM7;;*Apn >* compared to *met27,gce25k/FM7;;Apn >* ; see Fig. 3f). However, introducing hemyzygous *met27, gce25k* mutations (*met27, gce25k*/FM7 females) efficiently rescued the lethality and the size of *Apn > AdipoR-Ri* animals (Fig. 3f, g), indicative of a genetic interaction between AdipoR and JH signaling. Although a direct effect of brain AdipoR on peripheral JH targets is possible, our data suggest an antagonistic interaction between brain AdipoR signaling and JH production and signaling.

No difference was detected in *Akh* mRNA level or protein level in the RG of *Apn > AdipoR-Ri* larvae (Supplementary Fig. 3a, c), suggesting that AKH signaling is not majorly affected by AdipoR signaling. Moreover, APNs neuronal projections are distinct from those of AKH-producing cells (Supplementary Fig. 3b). Analysing ecdysone signaling, we found that expression of *E78*, an ecdysone receptor target, is delayed in *Apn > AdipoR-Ri* larvae (Supplementary Fig. 3d). This correlated with delayed pupariation (+12 h at 29 °C; Supplementary Fig. 3e) and the delayed accumulation of circulating and total levels of 20E (the active form of ecdysone). The same delays were observed in JHa-fed animals (Supplementary Fig. 3f–h). However, AdipoR is not present in PTTH-producing neurons (Supplementary Fig. 3i) and the mRNA levels of *ptth* are not altered in *Apn > AdipoR-Ri* larvae (Supplementary Fig. 3j). In addition, *Apn > AdipoR-Ri* larvae are significantly smaller than control animals, in contrast to *ptth* mutant larvae, which grow to larger size[19,20]. Finally, feeding *Apn > AdipoR-Ri* animals with 20E did not rescue their lethality (Supplementary Fig. 3k). These results indicate that while ecdysone production is defective in *Apn > AdipoR-Ri* larvae, this appears to be a consequence of increased JH production.

**Brain AdipoR regulates peripheral insulin sensitivity.** In light of the growth phenotypes, we examined the role of brain AdipoR in modulating peripheral insulin response. Using a *tubulin::GFP-PH* (*tGPH*) construct as an indicator of PI3-kinase activity[21], we observed a reduction of basal insulin signaling in the fat bodies of

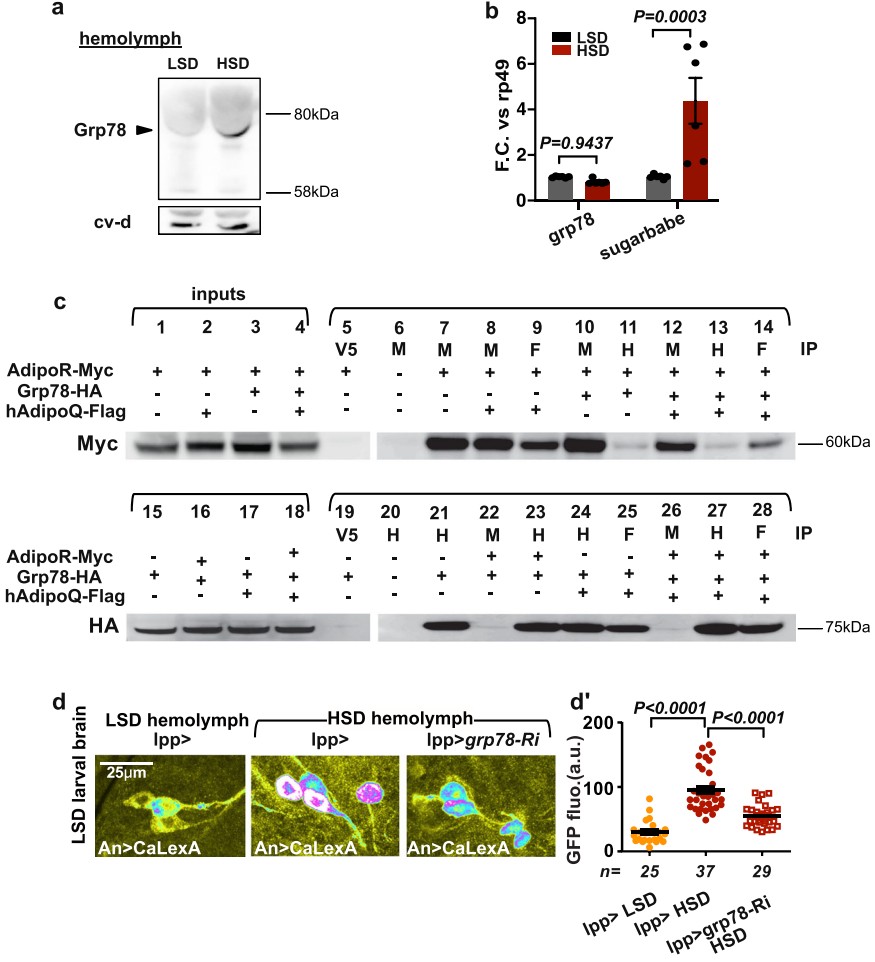

**Fig. 2 Grp78 is a sugar-induced inhibitory partner for AdipoR. a** Grp78 immunodetection in the hemolymph of low (LSD) and high (HSD) sugar diet-fed larvae. Cv-d is a control for hemolymph proteins. (the experiment was repeated three times independently with similar results). **b** *grp78* RNA levels in the whole larva in LSD (grey) versus HSD (red). qRT-PCR profiles, fold changes (FC) normalized to *rp49*. *sugarbabe* serves as a control for sugar induction. Statistical significance was tested using ordinary two-way ANOVA with Sidak's multiple comparisons test. n = 3 biologically independent replicates per experiment (two independent experiments). **c** Co-immunoprecipitations (co-IP) of AdipoR-Myc, Grp78-HA, and hAdipoQ-Flag expressed in S2 cells. Inputs (lanes 1–4; 15–18). Myc (M), HA (H), and Flag (F) IPs are used to visualize AdipoR-Myc (5–14) or Grp78-HA (19–28). (the experiment was repeated three times independently with similar results). **d** Brain from *Apn > CaLexA* larvae fed in LSD and incubated with the hemolymph from *lpp* > control larvae (LSD- n = 25, orange or HSD-fed n = 37, red points) or *lpp > grp78-Ri* larvae (−40%; HSD-fed n = 29, red squares). **d'** Quantification of GFP fluorescence in APN. n = independent averaged GFP intensity measurements in APN. Statistical significance was tested using a two-sided Mann–Whitney test. Data were presented as mean values ± SEM. P values are indicated in all panels. a.u. arbitrary units. Source data are provided as a Source Data file.

*Apn > AdipoR-Ri* larvae (Supplementary Fig. 4b). Then, we tested the ability of larval fat body explants to respond to human insulin in different genetic conditions. Compared to control explants, *Apn > AdipoR-Ri* fat bodies showed limited ability to activate PI3K after exposure to insulin (Fig. 4a, a')[22,23]. In addition, insulinemia, measured as circulating levels of *Drosophila* insulin-like peptide 2 (Dilp2) in hemolymph, was found increased in *Apn > AdipoR-Ri* and *Apn > AdipoR-TRIP* larvae (+55% and 19% respectively, Fig. 4c and Supplementary Fig. 4f). Conversely, the expression of an activated form of AdipoR (*AdipoR-act*) leads to a substantial decrease in circulating Dilp2 (Fig. 4c). Production of human AdipoQ in the larval fat body (*lpp > hAdipoQ*) lead to a similar decrease in insulinemia (Fig. 4d). This occurred without a change in *dilp2* transcription (Supplementary Fig. 4d). Notably, silencing *AdipoR* in the IPCs had no incidence on circulating Dilp2 levels (Supplementary Fig. 4e). When challenged for glucose tolerance (ref. [24] and Supplementary Materials for details), *Apn > AdipoR-RNAi* larvae did not clear glucose from hemolymph as efficiently as controls (Fig. 4f), confirming a state of insulin resistance. In line with this, mRNA levels for the

*Drosophila* insulin receptor gene (*Inr*) were found reduced in *Apn > AdipoR-Ri* larvae (Fig. 4g), as previously described in mouse models of insulin resistance[25].

**Insulin sensitivity is affected by JH signaling and Grp78.** Our finding that central AdipoR signaling controls peripheral JH targets suggested that circulating JH modulates insulin-response in peripheral tissues. Indeed, feeding larvae with JHa increased circulating Dilp2 levels compared to control-fed animals (Fig. 4e). In addition, when pretreated with JHa, fat body explants showed a reduced ability to respond to insulin stimulation (Fig. 4b and Supplementary Fig. 4c). In line with this, *Inr* transcription was reduced in fat body cells treated with JHa and did not vary significantly after insulin stimulation (Supplementary Fig. 4a). Therefore, JH signaling in peripheral tissues directly inhibits local insulin signaling. These observations are in line with the recent finding that Kr-h1, the transcriptional effector of JH, was found to bind to dFoxo and inhibits the transcription of several of its targets, including *Inr*[26]. Altogether, our data indicate that

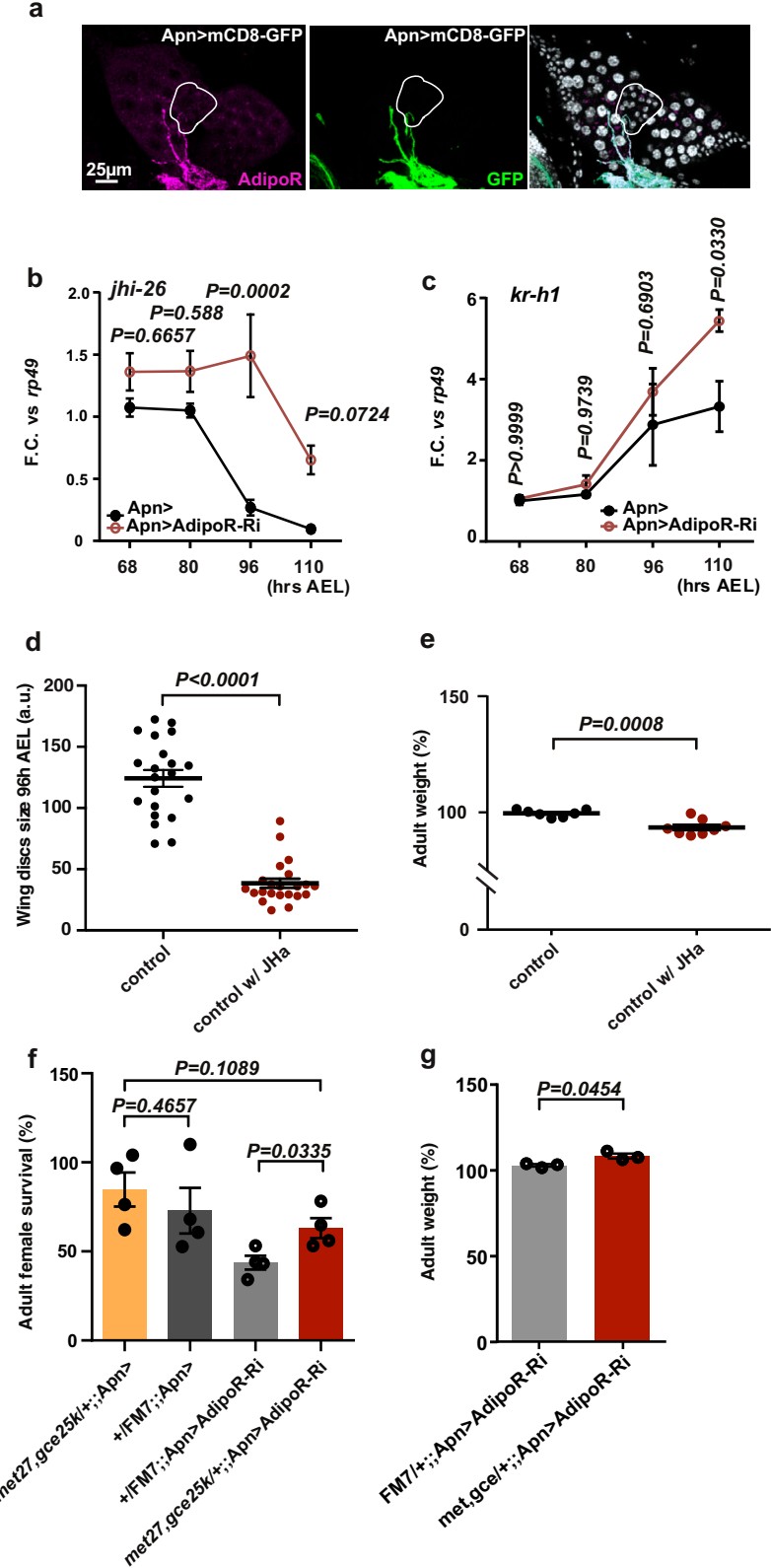

inhibition of adiponectin receptor signaling in the brain induces peripheral insulin resistance through an increase in JH signaling.

Finally, perturbations of *grp78* expression in fat cells (*lpp > grp78-Ri* and *lpp > grp78*) sensibly modified the levels of circulating Dilp2 (Fig. 4h, i), indicating that Grp78 acts as an adipose antagonist of AdipoR function and participates in peripheral insulin resistance in HSD conditions.

In conclusion, we describe a brain circuitry controlling peripheral insulin response, whereby a limited number of AdipoR-positive neurons regulate peripheral JH signaling, which in turn inhibit insulin response in peripheral organs. Moreover, we identify Grp78, a secreted protein previously identified in UPR response, as an antagonistic interactor of dAdipoR and its possible agonist ligands, produced upon high sugar feeding and

**Fig. 3 Central AdipoR function controls JH signaling. a** Anti-AdipoR and anti-GFP immunostaining of *Apn > mCD8-GFP* larval brains. Brain and ring gland are stained with DAPI, the corpora allata are circled. (*n* = 18 in five independent experiments). **b, c** *jhi-26* and *kr-h1* expression on whole larvae *Apn > AdipoR-Ri* (red) and *Apn >* (black). qRT-PCR profiles, fold changes (FC) normalized to *rp49*. Statistical significance was tested using ordinary two-way ANOVA with Sidak's multiple comparisons test. *n* = 2–3 biologically independent replicates per time point (three independent experiments). **d** Measure of JH analog (pyroproxifen)-feeding wing disc size. (control *n* = 22, black; control w/Jha *n* = 21, red, in three independent experiments). Statistical significance was tested using ordinary two-way ANOVA with Sidak's multiple comparisons test. **e** Measure of adult weight in JH analog feeding (control *n* = 7, black; control w/Jha *n* = 8, red). Statistical significance was tested using a two-sided unpaired *t*-test with Welch's correction. **f, g** Rescue of *Apn > AdipoR-Ri* female lethality (**f**) and adult weight (**g**) by *met27, gce25k* heterozygous mutations (*met27, gce25k/FM7; Apn > AdipoR-Ri* in red compared to *Apn > AdipoR-Ri* in light gray). *n* = 4 independent experiments (in **f**), *n* = 3 groups of 15 females of each genotype (in **g**). Statistical significance was tested using two-sided unpaired *t*-test with Welch's correction. Data were presented as mean values ± SEM. *P* values are indicated in all panels. a.u. arbitrary units. Source data are provided as a Source Data file.

participating in peripheral insulin resistance (Supplementary Fig. 4c). Some of the properties of JH in controlling developmental processes are shared with the vertebrate thyroid hormones[27]. Hyperthyroidism is associated with peripheral insulin resistance, suggesting mechanistic conservation of their metabolic functions. Finally, Grp78 is found in the blood of mice fed an HSD and heterozygous *grp78* mutant mice are protected from obesity when fed a hypercaloric diet[28], suggesting that central adiponectin signaling and Grp78 could play conserved roles in maintaining energy homeostasis in vertebrates.

## Methods

**Fly stocks and food**. The following *Gal4* lines were used in this study: *Apn-Gal4* (Janelia #48522, corresponding to a subregion of the promoter of the *Gycbeta100B* gene, referred to as *Apn-Gal4* in the text); *dilp2-Gal4* (BDSC#37516), *MHC-Gal4* (BDSC#55133), *ptc-Gal4* (BDSC#2017), *lpp-gal4*[29], *Akh-Gal4* (BDSC#25683), and *elav-Gal4* (BDSC#458). To invalidate *AdipoR*, two independent UAS-RNAi and a lethal EP insertion in the *AdipoR* gene were used: *UAS-AdipoR-RNAi*[GD] (called *UAS-AdipoR-Ri* in this study; VDRC#v40936), *UAS-AdipoR-RNAi*[TRIP] (called *UAS-AdipoR-TRIP* in this study; BDSC#67814), and *AdipoR-G6641* (called *Adi-poR*[M] in this study; BDSC#31800). Other lines used: *white1118* (control *w*); *UAS-mCD8GFP* (BDSC#5137), CaLexA system (*lexAop-CD8-GFP-2A-CD8-GFP; UAS-mLexA-VP16-NFAT, LexAop-rCD2-GFP*)[11]; *met27, gce25k* (gift from I. Miguel-Aliagua);*UAS-gpr78-RNAi* (VDRC#v101766); *UAS-grp78* (BDSC#5843), *UAS-sGFP*[30], tGPH (BDSC#8163). Lines used for circulating Dilp2 measurements were: *Dilp2HF* insertions combined or not with *elav-Gal4, lpp-Gal4, dilp2-Gal4,* or *Apn-Gal4* (original and combined *gd2HF* lines from refs. [31,32] and this study). *UAS-gfp-RNAi* (BDSC#9330) was used as a control for all the crosses and experiments (unless if notified).

Animals were reared on reference food (normal diet, ND) at 25 °C (or 29 °C) containing per liter: 10 g agar, 82,5 g corn flour, 60 g sugar, 34 g inactivated yeast extract, and 375 g Moldex (in Ethanol). LSD corresponds to the same without sugar and HSD to the same with 240 g sugar.

**Animal volume and weight measurements**. Flies were allowed to lay eggs 4 h on agar plates with yeast paste and first instar larvae were synchronized and transferred on the desired food under control conditions (30 L1/vial). Animals were allowed to develop until the stage of interest at 29 °C. Developmental timing, animal volume, adult weight, and larval imaginal disks/adult wing sizes were performed as described previously[33,34].

**Determination of larval and pupal volume**. Flies were allowed to lay eggs and first instar larvae were synchronized and transferred on the desired food under control conditions (30 L1/vial). Animals were allowed to develop until pupal stage at 29 °C. Pupal volume was measured using Fiji and calculated by using the formula: (4/3) π (L/2)(l/2) 2 (L length; l diameter).

**Plasmids and transgenic flies**. To generate transgenic *UAS-hAdipoQ* (Human Adiponectin), Adiponectin human cDNA (Origene, #TP315161) was PCR amplified and subcloned, using the Gateway cloning system (Invitrogen #K-2400-20), in Flag-tagged-UAS as destination vector (Carnegie Institution of Washington Vector collection). *UAS-AdipoR-Myc* was obtained from cloning a synthetic *AdipoR* cDNA (GeneSynthesis; GenScript) into a Myc-tagged-UAS as destination vector (Carnegie Institution of Washington Vector collection), using the Gateway cloning system (Invitrogen). A sequence of *AdipoR* corresponding to the cytoplasmic part of the receptor only (amino acids 1–201) was PCR amplified and subcloned in Myc-tagged-UAS as destination vector (Carnegie Institution of Washington Vector collection) to generate an activated form of *AdipoR* (*AdipoR-act*). *grp78* cDNA was subcloned from FlyORF clone (DGRC #GEO03341) to an HA-tagged-UAS vector as destination vector using the Gateway cloning system (Invitrogen; Carnegie

Institution of Washington Vector collection). Expression of the transgenes was validated by transfection in S2 cells and western blot. Constructs were then introduced into germ line by injections in the presence of the integrase (BestGene). (Primers sequences provided in Table S2; material available upon request).

**Adiponectin, JHa, and 20E feeding**. Human adiponectin recombinant protein (Merck, #SRP4901) was added to the food at a 5 ug/ml final concentration. Pyriproxifen (PESTANAL®, Merck, #34174) was used as JH analog (JHa, Ethanol stock at 1 mg/ml) at 2 ppm final in food used to reared animals (compared to control food without JHa). 20E (Sigma Aldrich, #5289-74-7) was added at 0.75 mg/ml to yeast paste during larval third instar on synchronized animals[35]. Dilution in PBS of a stock solution of 20E at 5 mg/ml in ethanol was used and ethanol was used as a control.

**Antibodies and immunohistochemistry**. To generate antisera against AdipoR protein (available upon request), two different couples of peptides were used as an immunogen in rats (doubleX program; Eurogentec). The chosen peptides correspond to amino acids KRRGWGPEDSLSPNDL (75–90)/LWDKFSEPALRPLRAG (321–334) and LQDNDFLHRGHRPPL (173–187)/VPERWFPGKFDIWGQ (388–402) of the sequence. Both are giving the same pattern. Wandering larvae were dissected in cold PBS on ice, fixed in 37% methanol-free formaldehyde (Polysciences Inc., #18814-10) for 25 min, washed several times in PBS + 0.1% Triton X-100 (PBT) before 2 h blocking in PBT containing 10% BSA. Primary antibodies were incubated overnight at 4 °C. The following primary antibodies were used: rat anti-AdipoR (1:400, this study); chicken anti-GFP (1:10000, Abcam #mAb13970; RRID:AB_300798), rabbit anti-Dilp5 (1:500, generated in Leopold's lab)[36], rabbit anti-Akh (1/500)[37], guinea pig anti-PTTH (1:400, generated in Leopold's lab)[38]. Alexa conjugated secondary antibodies were incubated for 2 h at room temperature then tissues of interest were mounted, after extensive washes, in SlowFade Gold antifade reagent (Molecular Probes, #S36936). The following secondary antibodies were used (1:400; Thermo Fisher Scientific): Alexa Fluor 488 goat anti-chicken (#A-11039), Alexa Fluor 546 goat anti-rat (#A-11081), Alexa Fluor 488 goat anti-rabbit (#A-11008), and Alexa Fluor 488 goat anti-guinea pig (#A-11073).

Fluorescence images were acquired by sequential scanning using a Leica SP5 DS (20× and 40× objectives) and processed with Fiji[39] Software.

**Antibody staining quantification**. For quantification of AdipoR staining in the APNs, stacks were taken for each cluster of *Apn* in each lobe from top to bottom limit of the staining. Slices of the whole stack were used to evaluate the fluorescence intensity using Fiji[39] (StagReg, Time Series Analyser V3.0, and Roi Manager plugins).

**NFAT quantification**. We used the calcium reporter *UAS-CaLexA* (calcium-dependent nuclear import of LexA, *lexAop-CD8-GFP-2A-CD8-GFP; UAS-mLexA-VP16-NFAT,* and *LexAop-rCD2-GFP*), allowing Ca++-dependent activation of *GFP* transcription[11]. For sugar tests, brains from *CalexA; Apn >* or *CaLexA; Apn >, >AdipoR-Ri* pre-wandering larvae (still in the food), grown on 0xS-1xS-4xS (LSD-ND-HSD) foods, were dissected on ice in PBS, and immunostaining was performed as described above. For hAdipoQ feeding, *CalexA; Apn >* animals grew on 4xS food supplemented with 5 ug/ml of recombinant human adiponectin (Merck, #SRP4901) were incubated overnight on a petri dish. Chicken anti-GFP antibody (1:10,000, Abcam #mAb13970) was used as primary.

For quantification, stacks were taken for each cluster of *Apn* in each lobe from top to bottom limit of the staining. Slices of the whole stack were used to evaluate the fluorescence intensity using Fiji[39] (StagReg, Time Series Analyser V3.0, and Roi Manager plugins). For each brain, one cluster was taken into account and three to four individual neurons per cluster were measured.

**Ex-vivo incubation**. Fat bodies from tGPH larvae were incubated with the desired compound or with control solutions (ethanol or water) diluted in HL6 medium[40], at room temperature: JHa (2 ppm for 30 min; Merck, #34174) or Insulin (0.5 uM

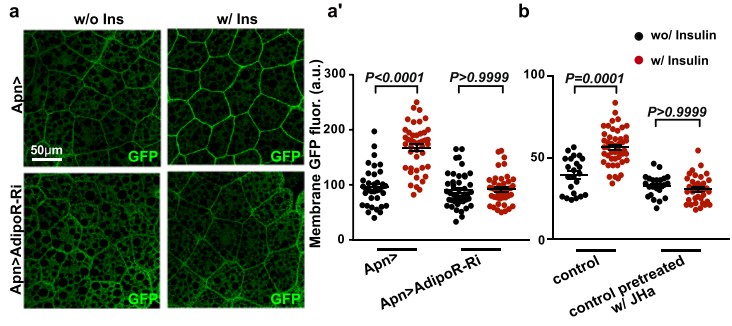

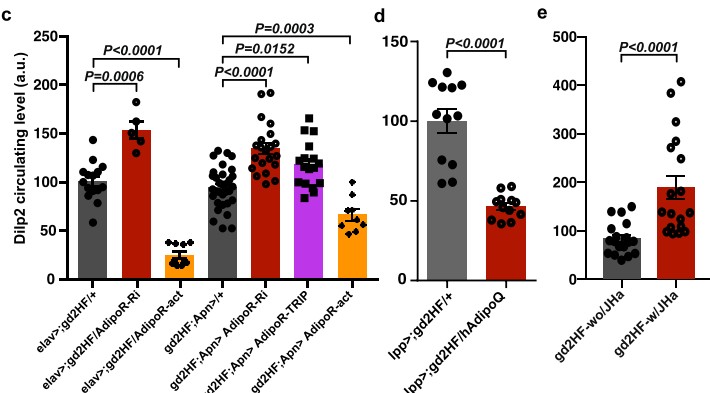

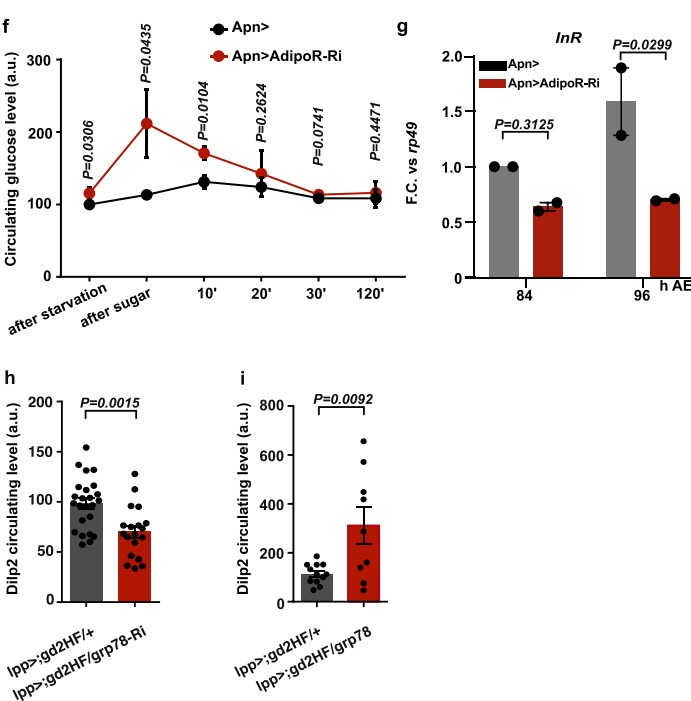

for 20 min; Merck, #i9278). For sequential incubations, after 30 min of incubation with JHa (or EtOH as a control), insulin (or H2O, as a control) was added for 30 min. The incubation mix was then removed and tissues were fixed and processed as described above. Chicken anti-GFP antibody (1:10,000, Abcam #mAb13970) was used as primary. Images were obtained with Leica SP5 (objective ×40) and processed with Fiji Software[39]. For quantification of fluorescence, an average of independent measurements per fat bodies of the indicated genotypes was performed.

Brains from *Apn > CalexA* larvae grown in 0xS were dissected in Schneider medium (Gibco™ #21020024) with FBS and incubated overnight at 18 °C with 20 ul of the hemolymph form control (*lpp >* ) or test (*lpp > grp78-Ri*) larvae grown in 0×S or 4×S. Tissues were then fixed and immunostaining was processed as described above with chicken anti-GFP antibody (see previously).

**S2 cells transfection and co-immunoprecipitation**. *UAS-AdipoR-Myc, UAS-hAdipoQ-Flag,* and/or *UAS-grp78-HA* were transfected in naïve S2 cells with the

**Fig. 4 Central AdipoR function and JH signaling control peripheral insulin sensitivity. a** *GFP* localization at the membrane in *tGPH* fat body explants −/+ incubation with human insulin, as a marker for insulin response. Fat bodies are stained with anti-GFP. (minimum *n* = 8 in two independent experiments). **a′** Quantification of membrane GFP staining in individual cells of larval fat bodies explants wo/insulin (black) or w/insulin incubation (red) from (**a**) *Apn* > wo/ins *n* = 36, w/ins *n* = 44; *Apn* > *AdipoR-Ri* wo/ins *n* = 44, w/ins *n* = 44). **b** Quantification of membrane GFP staining in individual cells of *tGPH* larval fat bodies explants pretreated or not with JHa before incubation with insulin (wo/insulin, black; w/insulin, red) (relates to Fig. S4c) (control wo/ins *n* = 23, w/ins *n* = 47; control wo/ins *n* = 23, w/ins *n* = 37). In (**a′**, **b**) *n* = number of averaged GFP intensity in fat bodies pieces. Statistical significance was tested using ordinary one-way ANOVA (Kruskal–Wallis test) with Dunn's multiple comparisons test. **c–e** Measure of hemolymph Dilp2 by ELISA in *elav* > ;*gd2HF*/+ (*n* = 15, gray), *elav* > ;*gd2HF/AdipoR-Ri* (*n* = 5, red), *elav* > ;*gd2HF/AdipoR-act* (*n* = 10, orange) in two independent experiments; *gd2HF;Apn* > /+ (*n* = 32, gray), *gd2HF;Apn* > /*AdipoR-Ri* (*n* = 21, red; four independent experiments), *gd2HF;Apn* > /*AdipoR-TRIP* (*n* = 17, purple; two independent experiments), and *gd2HF;Apn* > /*AdipoR-act* (*n* = 9, orange; three independent experiments); in *lpp* > ;*gd2HF* (*n* = 12, black) and *lpp* > ;*gd2HF/ hAdipoQ* (*n* = 12, red) in two independent experiments (in **c**, **d**), and upon feeding larvae with JHa during development *gd2HF-wo/Jha* (*n* = 18, black) and *gd2HF-w/Jha* (*n* = 18, red) in three independent experiments (in **e**). *n* = number of independent measurements. Statistical significance was tested using ordinary two-way ANOVA with Sidak's multiple comparisons test. **f** Larval glucose clearance after glucose feeding. Statistical significance at each time point was tested using an unpaired two-sided *t*-test. *n* = 3 independent experiments. **g** *InR* expression on whole larvae (qRT-PCR profiles, fold changes (FC) normalized to *rp49*). Statistical significance was tested using ordinary two-way ANOVA with Sidak's multiple comparisons test. *n* = 3 biologically independent replicates per experiment (two independent experiments). **h**, **i** Measure of hemolymph Dilp2 by ELISA in *lpp* > ;*gd2HF/grp78-Ri* (*n* = 19, red) and *lpp* > ;*gd2HF/grp78* (*n* = 9, red) compared to *lpp* > ;*gd2HF* (*n* = 24 and *n* = 11, respectively, black) in four and three independent experiments (**h** and **i**, respectively) *n* = number of independent averaged measurements. Statistical significance was tested using ordinary two-way ANOVA with Sidak's multiple comparisons test. Data were presented as mean values ± SEM. *P* values are indicated in all panels. a.u. arbitrary units. Source data are provided as a Source Data file.

pAct-GAL4 vector using Lipofectamine reagent kit (Invitrogen #18324012) in a proportion of 1:1:1. Lysis and immunoprecipitation were performed as described previously[41].

**Western blot.** Hemolymph samples from 96 h AEL larvae of the indicated genotypes, reared on 0×S (LSD) and 4×S (HSD) foods, were extracted as described in ref. [42], and prepared for western blotting. For control (*lpp* > *GFP-RNAi*; Fig. 2a) and sGFP expressing larvae (*lpp* > *sGFP*, Fig. S2d), the experiment was performed on 10 ug of proteins. For *lpp* > *grp78-RNAi*, the experiment was performed on 5 ul of pure larval hemolymph, the low amount of total serum proteins preventing the measurement of protein amount by Bradford. In both cases, cv-d was used as a normalizer for hemolymph proteins[43].

Proteins (from IP and hemolymph) were resolved by SDS-PAGE, using 4 – 12% gradient gels (NuPage Novex Gel, Invitrogen #NP0335)[41]. Primary antibodies used were: rabbit anti-Grp78 (1:1000, Novus Biologicals #NBP1-06274), rabbit Anti-GFP N-ter (1:1000, Merck, #G1544), guinea-pig anti-Cv-d (1:2000[43], gift from Eaton lab); mouse anti-Myc 9E10 (1:1000, Santa Cruz Biotechnology, #sc-40), rabbit anti-Myc A-14 (1:1000, Santa Cruz Biotechnology, #sc-789), rat anti-HA (1:1000, Merck/Roche, #ROAHAHA), and mouse anti-FLAG® M2 (1:1000, Merck, F3165). Membranes were washed in PBS tween 0.1% and incubated with secondary antibodies in this buffer for 2 h at room temperature. The following HRP conjugated secondary antibodies were used (Thermo Fisher Scientific): Goat anti-Rabbit (1:2500, #G-21234), Goat anti-mouse (1:2500, #G-21040), and anti-guinea pig (1:2500, #A18769). Chemiluminescence was observed using the ECL detection system (Pierce). Images were generated using Fiji[39].

**Glucose tolerance measurements.** Seventy-two hours AED larvae were starved for 2 h on a humid petri dish plated with agarose and immerged in a petri dish containing a saturated sugar solution for 45 min. Larvae are removed from sugar, washed, dried, and put back on a humid petri dish for a challenging time. For each point, hemolymph of three times eight larvae (in triplicates/genotype) was picked on ice and submitted to total glucose measurement as described in ref. [44], using the GAGO-20 kit from Merck (according to the manufacturer's instructions). Absorbance at 540 nm was measured using a TECAN microplate reader.

**Dilp2HF Elisa.** Circulating Dilp2 was quantified by sandwich Elisa as described in ref. [31]. Almost three times 1 ul of clean hemolymph was picked from groups of ten larvae of the indicated genotypes (one biological replicate) or fed with 2 ppm JHa during all development and submitted, to Elisa test. The relative ratio of tested animals is given relative to control animals, normalized to the volume of hemolymph.

**Ecdysone measurements.** Five Larvae were homogenized and extracted in 250 ul methanol or 5 ul of hemolymph in 200 ul methanol. Extractions were then submitted to a competitive ELISA test as previously published[45] to evaluate the amount of total or circulating 20E, using the detection kit from SpiBio (Bertin reagents #A05120) as recommended by the manufacturer. Absorbance at 415 nm was detected using a TECAN microplate reader.

**qRT-PCR.** Whole larvae (all figures panels except Fig. S4a) or dissected tissues (Fig. S4a) were collected, washed in PBS 1×, and dried, then frozen in liquid nitrogen. Total RNA was extracted using Qiagen RNeasy lipid tissue mini kit according to the manufacturer protocol (#74804). RNA samples (3 μg per reaction) were treated with DNase and reverse-transcribed using SuperScript II reverse transcriptase (Invitrogen #18064014) and the generated cDNA was used for real-time RT-PCR (StepOne Plus, Applied Biosystem) using PowerSYBRGreen PCR master mix (Applied Biosystem #4367659) as previously described[46]. Three biological replicates were collected for each sample and technical triplicate was conducted for each (Primers sequences are provided in Table S2).

**DTT and tunicamycin treatment.** Control larvae were allowed to grow either on LSD or HSD diet until pre-wandering. For DTT treatment, three groups of ten LSD-fed larvae were starved 2 h and incubated for 4 h in Schneider's medium ± 5 mM DTT (Merck #10197777001). Larvae were then washed and frozen in liquid nitrogen. Three groups of ten LSD- or HSD-fed larvae were washed with PBS and frozen simultaneously. RNA extraction and qRT-PCR were performed as described above. Independent experiments were performed using three biological replicates tested in triplicates. For tunicamycin treatment, fat tissues from *lpp* > *xbp1-GFP* larvae were incubated 1 h either with 10 ug/ml tunicamycin (1:1000 dilution of 10 mg/ml stock solution, Sigma Aldrich #T7765) or with 1:1000 DMSO in Schneider's medium at room temperature. Tissues were then washed with PBS and fixed in 37% methanol-free formaldehyde (Thermo Fisher Scientific, #28906). Anti-GFP immunostaining was performed as described above.

**Mass spectrometry analysis.** Ubiquitously expressed Myc-AdipoR protein was immunoprecipitated from larvae using magnetic microparticles (MACS purification system, Miltenyi Biotech) according to the manufacturer's instructions and as previously described[47]. μMACS magnetic microbeads are coated with a monoclonal anti-Myc antibody (*Miltenyi* Biotec). Negative controls, wild-type larvae, and the negative control protein myc-SMO (kindly provided by Dr L. Ruel) were purified by affinity with the same conditions to remove unspecific proteins. A positive control was carried out with a co-expression of Myc-AdipoR with hAdipoQ. Co-immunoprecipitation experiments were carried out in triplicates. Proteins were eluted out of the magnetic stand with the SDS-loading buffer from the kit.

Eluted proteins were digested with sequencing-grade trypsin (Promega) and analyzed by nanoLC-MS/MS on a QExactive+ mass spectrometer coupled to an EASY-nanoLC-1000 (Thermo Fisher Scientific) as described previously[48]. Data were searched against the Flybase database (release r6.16) with a decoy strategy. Peptides were identified with Mascot algorithm (version 2.5, Matrix Science, London, UK) and data were further imported into Proline 1.4 software (http://proline.profiproteomics.fr/). Proteins were validated on Mascot pretty rank equal to 1, and 1% FDR on both peptide spectrum matches (PSM score) and protein sets (protein set score). The total number of MS/MS fragmentation spectra was used to quantify each protein from at least three independent biological replicates. This spectral count was submitted to a negative-binomial test using an edgeR GLM regression through R (R v3.2.5). For each identified protein, an adjusted *P* value corrected by Benjamini–Hochberg was calculated, as well as a protein fold change (FC). We used a in-house developped R script, called IPinquiry, freely available under Github (https://github.com/hzuber67/IPinquiry4).

**RNAi screen of candidate genes**. Candidate genes were selected on the enrichment value of the MS results. A blind RNAi screen was performed using two independent lines for each candidate (VDRC KK/GD and TRIP, if available). The RNAi were expressed with *ppl* > (expressed in the gut and the fat body, both tissues assumed to be the two most important secretory tissues of the larva). Pupal lethality was checked as a read-out.

Top candidate genes of the MS were screened in priority and among them, ones known to encode secreted or putative secreted proteins. A non-exhaustive list of candidates is presented in Fig. S2c.

**Statistical analysis and reproducibility**. Statistical analyses were performed with Graphpad Prism V9. The statistical test used for each experiment and the *P* values are indicated in the corresponding Figure and Figure legends. Experiment replications are summarized in Table S1.

**Software**. Mascot Algorithm v2.5 (Matrix Science, London, UK) and Proline software v1.4 (http://proline.profiproteomics.fr) were used for the mass spectrometry.

GraphPad Prism V9 was used for statistical analysis and graphical representation of the data. Fiji (ImageJ version 2.0.0-RC-69/1.52n) was used for Image treatment from microscopy and western blot. Affinity Photo v.1.8.5 and Affinity Designer v.1.9.3 (Serif, Europe) were used for Figures elaboration.

## Data availability

The processed mass spectrometry data are available at PRIDE repository: ProteomeXchange accession: PXD013689

Project Webpage: http://www.ebi.ac.uk/pride/archive/projects/PXD013689

FTP Download: ftp://ftp.pride.ebi.ac.uk/pride/data/archive/2021/06/PXD013689/

Source data are provided with this paper.

## Code availability

The custom R script used for mass spectrometry analysis is called IPinquiry and is freely available from Github (https://github.com/hzuber67/IPinquiry4).

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

## Acknowledgements

We thank T. Pihl for help with the experiments and laboratory members for discussions; We thank I. Miguel-Aliaga, K. Masuyama, the Bloomington Stock Center, and the Vienna Drosophila RNAi Center for fly stocks. Funding is from Inserm, CNRS, the Fondation ARC pour la recherche sur le cancer (grant no PGA120150202355 to P.L.), the European Research Council (ERC Advanced grant no 268813 to P.L.).

## Author contributions

Conceptualization and methodology: N.A., M.B., and P.L. Experimental investigation: N.A. and M.B. except mass spectrometry: L.K. and P.H. Formal analysis: N.A., Writing an original draft, supervision, and project administration: N.A. and P.L. Visualization: N.A. Funding acquisition: P.L.

## Competing interests

The authors declare no competing interests.
