## [Transparent Peer Review File · Nature Communications]

REVIEWER COMMENTS

Reviewer #1 (Remarks to the Author):

The authors have nicely addressed the issues raised in my original review. They have included a large amount of new data which have significantly strengthened the manuscript.

Reviewer #2 (Remarks to the Author):

The authors have done several things that have really improve the revised manuscript. I have a few comments below though:

I appreciate the authors response, but it is still not completely clear to me whether the Adiponectin receptor Positive Neurons (Apn)-Gal4 express exclusively in neurons or also in peripheral tissues, which could contribute to some of the effects? Perhaps the authors could use another neuronal driver such as nSyb-Gal4 to show that effects can be contributed to the role of AdipoR in the nervous system. While I think it should be up to the authors, I believe that the work could also be further strengthened by using another GAL4 driver lines such as a T2A::GAL4 insertion on the native AdipoR locus. That could be used both to visualize expression and to knock AdipoR down.

To strengthen the link between the APNs and JH signaling, the authors have now done a rescue experiments with met, gce mutants. This shows that loss of met/gce signaling, and thus JH signaling, increases adult size of animals with AdipoR knockdown in the APNs. While this is a good experiment, the authors should show that it rescues the larval growth rate. Increased adult size does not necessarily have reflect altered systemic growth during development. It could also indicate increased adiposity or other things. Larval growth can be analyzed using FM7-GFP as a balancer to distinguish FM7 is the larval stage. Preferably the experiment should also include met, gce alone as well as a wild type control, to see whether met/gce loss alone increases size, which could indicate that effects are merely additive.

Some other minor suggestions:

The authors state that knockdown of AdipoR reduces larval growth rate referring to figure 1e. While larvae with knockdown of AdipoR appear smaller 96 h AEL, these larvae are similar in size or even bigger than controls 120 h AEL. That would suggest that larvae with reduced AdipoR expression grow faster and not slower as stated.

The authors say that expression of AdipoR or human adiponectin induces systemic growth based on the increased adult weight. However, these adults could be bigger (as mentioned above) because they store more fat or retain water. I think it would be better to look at larval growth rate to support that it induces systemic growth.

Figure 1I. The data seem to suggest that the hypomorphic mutants are small, but grow as fast as controls, or even faster, from 96 h to 120 h AEL. However, the authors say they grow slower.

The authors perform some nice ex vivo experiments to show that Grp78 is produced by the fat tissue and acts antagonist of AdipoR. Brains cultured in hemolymph from animals with Grp78 knockdown does not increase APN activity. Is it possible that Grp78 knockdown affects the release of other adipokines from the fat tissue, which could explain these effects?

The APNs projects to the ring gland and the authors does a lot of work to show that it affects JH signaling from the CA part of this glands. However, it looks like the APN projections terminate close to the CC part of the ring gland which produces adipokinetic hormone (AKH). AKH promotes insulin signaling in response to dietary sugar, and thereby stimulates systemic growth, by regulating DILP3 1. The authors have addressed this by showing that AdipoR knockdown does not affect Akh transcripts or protein levels. However, it would be better to quantify AKH to support the representative image in Fig. S3c to further exclude contribution of AKH to the phenotypes they observed.

1 Kim, J. & Neufeld, T. P. Dietary sugar promotes systemic TOR activation in Drosophila through AKH-dependent selective secretion of Dilp3. Nature communications 6, 6846, doi:10.1038/ncomms7846 (2015).

Reviewer #3 (Remarks to the Author):

The authors have dealt with many, but not all, of the major criticisms of the three reviewers and the manuscript has been improved accordingly. They addressed well several of my own criticisms, including one of the major concerns that Grp78 may not be a specific ligand for AdipoR. This unsubstantiated claim has now been removed. Indeed, as the authors now imply in their rebuttal, the new co-IP data that human AdipoQ binds to fly AdipoR may be consistent with the revised interpretation. In addition, new experiments have been conducted that now rule out that the HSD induced GRP78 mechanism involves one of the branches of the UPR (IRE/XBP1) or an effect upon general secretion.

However, there are some remaining worries, especially about whether the data support the major conclusions about the mechanism of action of JH.

1) The conclusion that central AdipoR function controls peripheral insulin sensitivity through JH signaling does not seem to be well supported. Much of the data for this major point are in revised Fig 4, which does show that JH feeding regulates insulin levels (Dilp2 in hemolymph) and insulin-stimulated fat body PI3K signaling (tGPH). However, the word "through" in the title may or may not be correct here. In other words, data presented in this figure do not allow the authors to distinguish whether JH acts between AdipoR and insulin signalling/sensitivity or, alternatively, whether JH might act in a parallel pathway impinging upon insulin signalling/sensitivity. What seems to be missing are clear experiments showing whether or not AdipoR regulates JH production/signalling. These are essential to provide the necessary evidence supporting major conclusions about JH in this manuscript and in the abstract: "JH serves as an effector for brain AdipoR signaling". This is also necessary to support the arrows to and from JH,

depicted in the working model (Fig S4g).

2) In several revised figures, the data is now substantially different than in the previous version of the manuscript, casting doubt on the reproducibility of the results. For example:

-Why has the sugarbabe fold change halved in Fig 2b the revised compared to previous manuscript?

-Why has % adult female survival in Fig 3f more than doubled in the revised compared to previous manuscript? e.g FM7 Apn driver controls go from to ~40% to ~85%.

3) This leads me to repeat my previous request for the authors to specify the number of independent experiments (i.e done on different days) that have been conducted for each figure. This would strengthen those cases where one figure carries a lot of mechanistic weight such as Fig 2d' - the key data supporting the major conclusion that Grp78 is needed in fat cells to modulate brain APN activity.

REVIEWER COMMENTS

Reviewer #1 (Remarks to the Author):

The authors have nicely addressed the issues raised in my original review. They have included a large amount of new data which have significantly strengthened the manuscript.

Reviewer #2 (Remarks to the Author):

The authors have done several things that have really improve the revised manuscript. I have a few comments below though:

I appreciate the authors response, but it is still not completely clear to me whether the Adiponectin receptor Positive Neurons (*Apn*)-Gal4 express exclusively in neurons or also in peripheral tissues, which could contribute to some of the effects? Perhaps the authors could use another neuronal driver such as *nSyb*-Gal4 to show that effects can be contributed to the role of AdipoR in the nervous system. While I think it should be up to the authors, I believe that the work could also be further strengthened by using another GAL4 driver lines such as a T2A::GAL4 insertion on the native AdipoR locus. That could be used both to visualize expression and to knock AdipoR down.

We understand the concerns of the reviewer, which we attempted to addressed in our previous revision. Indeed, we previously provided data showing that neither fat body nor muscle drivers recapitulate the effects observed with *Apn*> *AdipoR-RNAi* (in these cases, we rather observe a faint increase in body weight, see Fig. S1l).

We now use a combination of *Apn-Gal4*>*UAS-GFP* to label specifically all cells and tissues targeted by the *Apn*> driver. This experiment shows no peripheral larval tissue targeted in addition to brain neurons (Fig. S1e).

In support of this, we show in Fig. 4c that *elav*>*AdipoR-RNAi* recapitulates the effect on insulinemia observed with *Apn*>*AdipoR-RNAi*.

To strengthen the link between the APNs and JH signaling, the authors have now done a rescue experiments with *met*, *gce* mutants. This shows that loss of *met/gce* signaling, and thus JH signaling, increases adult size of animals with AdipoR knockdown in the APNs. While this is a good experiment, the authors should show that it rescues the larval growth rate. Increased adult size does not necessarily have reflect altered systemic growth during development. It could also indicate increased adiposity or other things. Larval growth can be analyzed using FM7-GFP as a balancer to distinguish FM7 is the larval stage. Preferably the experiment should also include *met*, *gce* alone as well as a wild type control, to see whether *met/gce* loss alone increases size, which could indicate that effects are merely additive.

While we understand the reviewer's request and appreciate the relevance of the proposed experiment, the genetic crosses to establish these new lines and the experiments would take 3 months. This is therefore not compatible with the timing of our revisions. However, we point out the fact that the rescue of weight and lethality with *met,gce* mutants is further supported by the analysis of JH targets by qRT-PCR in *Apn*>*AdipoR-RNAi* conditions (Fig. 3b,c).

Some other minor suggestions:

The authors state that knockdown of AdipoR reduces larval growth rate referring to figure 1e. While larvae with knockdown of AdipoR appear smaller 96 h AEL, these larvae are similar in size or even bigger than controls 120 h AEL. That would suggest that larvae with reduced AdipoR expression grow faster and not slower as stated.

We guess the author refers to ex- Fig.S1e, now Fig.S1f.

AdipoR-RNAi larvae are smaller both at 72hr and 96hr, indicating slower growth. This is consistent with smaller wing discs observed at 96hr (Fig. 1b). Indeed *AdipoR-RNAi* larvae seem to catch up at 120hr. However, these animals are delayed by 10hr at pupariation, which implies that the final volume retraction that takes place in *wt* animals at 120hr is not yet seen in the *AdipoR-RNAi* larvae. This is responsible for the apparent catch up in volume compared to *wt*.

The authors say that expression of AdipoR or human adiponectin induces systemic growth based on the increased adult weight. However, these adults could be bigger (as mentioned above) because they store more fat or retain water. I think it would be better to look at larval growth rate to support that it induces systemic growth.

Apn>AdipoQ or *>AdipoR-act* adults do not show a bloating phenotype, which could induce increased volumes despite reduced dry mass. Unfortunately we do not have data concerning larval growth rate in *Apn>AdipoQ* or *>AdipoR-act* conditions. However we provide a series of converging evidences that make us confident that larval growth is affected by AdipoR function (see Fig. 1b,c,d; Fig. S1f,g,h).

Figure 1I. The data seem to suggest that the hypomorphic mutants are small, but grow as fast as controls, or even faster, from 96 h to 120 h AEL. However, the authors say they grow slower. See our answer to minor point 1: at 120hr, *wt* larvae start retracting, while *AdipoR^M* mutants are slightly delayed, therefore body retraction has not started yet.

The authors perform some nice ex vivo experiments to show that Grp78 is produced by the fat tissue and acts antagonist of AdipoR. Brains cultured in hemolymph from animals with Grp78 knockdown does not increase APN activity. Is it possible that Grp78 knockdown affects the release of other adipokines from the fat tissue, which could explain these effects?

We cannot exclude this possibility, however, we show that Grp-78 is secreted and binds to AdipoR, suggesting that it could directly affect Apn neurons. Grp78 could also modulate the action of a potential endogenous agonist of AdipoR, as stated in the text.

The APNs projects to the ring gland and the authors does a lot of work to show that it affects JH signaling from the CA part of this glands. However, it looks like the APN projections terminate close to the CC part of the ring gland which produces adipokinetic hormone (AKH). AKH promotes insulin signaling in response to dietary sugar, and thereby stimulates systemic growth, by regulating DILP3¹. The authors have addressed this by showing that AdipoR knockdown does not affect Akh transcripts or protein levels. However, it would be better to quantify AKH to support the representative image in Fig. S3c to further exclude contribution of AKH to the phenotypes they observed.

¹Kim, J. & Neufeld, T. P. Dietary sugar promotes systemic TOR activation in *Drosophila* through AKH-dependent selective secretion of Dilp3. *Nature communications* 6, 6846, doi:10.1038/ncomms7846 (2015).

We thank the reviewer to point this interesting paper. In this paper, Kim et al. propose a mechanism for the **activation of IIS** specifically in response to trehalose. Here we show that high sucrose feeding leads to a **suppression of IIS** through activation of Apns and JH-induces insulin resistance. So we do not think that the mechanism described by Kim et al. could account for what we observe.

Reviewer #3 (Remarks to the Author):

The authors have dealt with many, but not all, of the major criticisms of the three reviewers and the manuscript has been improved accordingly. They addressed well several of my own criticisms, including one of the major concerns that Grp78 may not be a specific ligand for AdipoR. This unsubstantiated claim has now been removed. Indeed, as the authors now imply in their rebuttal, the new co-IP data that human AdipoQ binds to fly AdipoR may be consistent with the revised

interpretation. In addition, new experiments have been conducted that now rule out that the HSD induced GRP78 mechanism involves one of the branches of the UPR (IRE/XBP1) or an effect upon general secretion.

However, there are some remaining worries, especially about whether the data support the major conclusions about the mechanism of action of JH.

1) The conclusion that central AdipoR function controls peripheral insulin sensitivity through JH signaling does not seem to be well supported. Much of the data for this major point are in revised Fig 4, which does show that JH feeding regulates insulin levels (Dilp2 in hemolymph) and insulin-stimulated fat body PI3K signaling (tGPH). However, the word "through" in the title may or may not be correct here. In other words, data presented in this figure do not allow the authors to distinguish whether JH acts between AdipoR and insulin signalling/sensitivity or, alternatively, whether JH might act in a parallel pathway impinging upon insulin signalling/sensitivity.

We agree with the Reviewer concerning Fig. 4 and have modified the title as follows: "Central AdipoR function and JH signaling control peripheral insulin sensitivity".

What seems to be missing are clear experiments showing whether or not AdipoR regulates JH production/signalling. These are essential to provide the necessary evidence supporting major conclusions about JH in this manuscript and in the abstract: "JH serves as an effector for brain AdipoR signaling". This is also necessary to support the arrows to and from JH, depicted in the working model (Fig S4g).

We have attempted to measure JH circulating levels in *Apn>AdipoR-RNAi* conditions by mass spectrometry, but this turned out to be technically challenging and no consistent data was obtained. However, we believe we establish a logical link between AdipoR and JH according to the following data:

-Fig. 4a,a': we establish a positive link between AdipoR signaling and peripheral IIS. **AdipoR → IIS**

- Fig. 3b,c: we show that AdipoR signaling affects systemic JH signaling: when AdipoR is silenced in *Apns*, JH signaling is increased. Therefore, AdipoR is genetically upstream of JH signaling (we made an error in the color code for Fig. 3c, we have now corrected this error). **AdipoR — I JH signalling**

- Fig. 3f,g: we demonstrate a genetic interaction between AdipoR and JH pathways.

From there, we conclude that AdipoR function antagonizes JH signaling. **AdipoR — I JH signalling**

- Fig. 4a',b,c,e, Fig. S4a,c : we show that JH feeding recapitulates all effects of AdipoR silencing on IIS. **JH signaling — I IIS**

From there, we propose that AdipoR exerts its function on IIS through a modulation of JH signaling.

AdipoR — I JH signaling — I IIS

2) In several revised figures, the data is now substantially different than in the previous version of the manuscript, casting doubt on the reproducibility of the results. For example:

-Why has the *sugarbabe* fold change halved in Fig 2b the revised compared to previous manuscript?

We apologize for the mistake, which appeared during the conversion of our figure into GraphPad.

We have restored it in its initial version (indeed showing a 4x increase in *sugarbabe* induction).

-Why has % adult female survival in Fig 3f more than doubled in the revised compared to previous manuscript? e.g. FM7 *Apn* driver controls go from ~40% to ~85%.

This experiment was repeated after a remark from Rev1 about the survival rates in our controls. Here is the response that was made to the reviewer:

"We thank the reviewer for pointing this discrepancy to us. We have repeated these experiments and we do not observe significant differences in survival rate between the two controls

(*met27,gce25k/+;;Apn* versus *+ /FM7;;Apn*), see new Fig. 3f." For this new set of experiments, more

than 1000 larvae were used and each dot on the graph represents the result of one experiment. The food conditions were improved to increase survival of our control animals.

3) This leads me to repeat my previous request for the authors to specify the number of independent experiments (i.e done on different days) that have been conducted for each figure. This would strengthen those cases where one figure carries a lot of mechanistic weight such as Fig 2d' - the key data supporting the major conclusion that Grp78 is needed in fat cells to modulate brain APN activity.

We deeply apologize for the confusion. Indeed, we prepared a revised version with this information provided in figure legends. By mistake, an earlier version was submitted. We have now corrected this error. We also added a Supplementary table 1 recapitulating all data points and number of independent experiments used for each figure in the paper.

REVIEWER COMMENTS

Reviewer #2 (Remarks to the Author):

I find that the authors' response and revision have addressed the concerns and comments. It looks ready for publication.

Reviewer #3 (Remarks to the Author):

The authors have satisfactorily addressed my second criticism but the first and third ones still remain an issue.

1) The conclusion that central AdipoR function controls peripheral insulin sensitivity through JH signaling does not seem to be well supported.

*It is welcome that the authors agree with this criticism as it applies to Fig4 and have corrected the figure title accordingly. Although it is unfortunate that the MS measurements of JH did not yield consistent data supporting AdipoR regulation of JH levels, I do understand that these assays can be tricky to get set up. That leaves us with the list of points they describe in the response letter as the only evidence for a logical link between AdipoR and JH. Unfortunately, my original criticism still stands as the evidence in this list does not convincingly demonstrate that JH functions inbetween AdipoR and IIS. This is because the finding that AdipoR knockdown increases jhi-26 and kr-h1 mRNA expression in whole larvae (Fig3b,c) does not provide strong enough evidence that AdipoR acts via/through JH signalling to regulate IIS. As I implied before, AdipoR may not act upstream of JH production but rather in parallel to regulate jhi-26 and kr-h1, which would also be consistent with the genetic interactions observed in and the genetic interactions in Fig3f,g. This alternative possibility is at odds with the revised manuscript title, abstract, conclusions and model in Fig S4g and therefore the paper is not future proof.

2) In several revised figures, the data is now substantially different than in the previous version of the manuscript, casting doubt on the reproducibility of the results.

*This criticism has been addressed and the authors have now corrected Fig2b and addressed the discrepancy in the Fig 3F controls.

3) This leads me to repeat my previous request for the authors to specify the number of independent experiments (i.e done on different days) that have been conducted for each figure. This would strengthen those cases where one figure carries a lot of mechanistic weight such as Fig 2d' - the key data supporting the major conclusion that Grp78 is needed in fat cells to modulate brain APN activity.

*The additional information on n numbers in the figure legends and in Supplementary Table 1 is an improvement. However, it appears that the main criticism about the need to specify the number of

independent experiments (i.e done on different days) still stands-with the exception of two panels where it is now clear (Fig4f and S3f-h). This is because many of the figure panels listed in Supplementary Table 1 do not clearly specify the number of independent experiments (i.e done on different days), instead they talk about other things. Two examples are “number of samples measured” (Fig 1b,1c, 3d, S1g) and “total number of biological replicates for each genotype measured in independent experiments” (4c,e and 4h,i and S4e,f). In the former case, we are left wondering whether this is only one independent experiment -which would not generally be acceptable. In the latter case, we are wondering whether there are multiple n numbers from each independent experiment and several independent experiments all lumped together into one population. If this is the case, then good statistical practice would be to state clearly how many independent experiments were done and then to use an alternative statistical method that accounts for the different variances within and between experiments, such as mixed-effect models.

Reviewer #2 (Remarks to the Author):

I find that the authors' response and revision have addressed the concerns and comments. It looks ready for publication.

Reviewer #3 (Remarks to the Author):

The authors have satisfactorily addressed my second criticism but the first and third ones still remain an issue.

1) The conclusion that central AdipoR function controls peripheral insulin sensitivity through JH signaling does not seem to be well supported.

*It is welcome that the authors agree with this criticism as it applies to Fig4 and have corrected the figure title accordingly. Although it is unfortunate that the MS measurements of JH did not yield consistent data supporting AdipoR regulation of JH levels, I do understand that these assays can be tricky to get set up. That leaves us with the list of points they describe in the response letter as the only evidence for a logical link between AdipoR and JH. Unfortunately, my original criticism still stands as the evidence in this list does not convincingly demonstrate that JH functions inbetween AdipoR and IIS. This is because the finding that AdipoR knockdown increases jhi-26 and kr-h1 mRNA expression in whole larvae (Fig3b,c) does not provide strong enough evidence that AdipoR acts via/through JH signalling to regulate IIS. As I implied before, AdipoR may not act upstream of JH production but rather in parallel to regulate jhi-26 and kr-h1, which would also be consistent with the genetic interactions observed in and the genetic interactions in Fig3f,g. This alternative possibility is at odds with the revised manuscript title, abstract, conclusions and model in Fig S4g and therefore the paper is not future proof.

To positively answer this comment, we have modified the title of the manuscript by removing the reference to a direct role of AdipoR on JH. We also modified the abstract accordingly. Finally, we mention in the text the possibility that AdipoR signaling could act directly to regulate JHi-26 and Kr-H1. We also modified the text by removing all mentions to "JH production" and replacing it by "JH signaling", referring to the effect on Kr-h1, the transcriptional effector of JH.

Concerning Fig. S4g, this is a proposed working model, as mentioned in the legend. However we also changed the legend and removed the mention of AdipoR acting on JH production, replacing it by acting on JH signaling.

2) In several revised figures, the data is now substantially different than in the previous version of the manuscript, casting doubt on the reproducibility of the results.

*This criticism has been addressed and the authors have now corrected Fig2b and addressed the discrepancy in the Fig 3F controls.

3) This leads me to repeat my previous request for the authors to specify the number of independent experiments (i.e done on different days) that have been conducted for each figure. This would strengthen those cases where one figure carries a lot of mechanistic weight such as Fig 2d' - the key data supporting the major conclusion that Grp78 is needed in fat cells to modulate brain APN activity.

*The additional information on n numbers in the figure legends and in Supplementary Table 1 is an improvement. However, it appears that the main criticism about the need to specify the number of

independent experiments (i.e done on different days) still stands-with the exception of two panels where it is now clear (Fig4f and S3f-h). This is because many of the figure panels listed in Supplementary Table 1 do not clearly specify the number of independent experiments (i.e done on different days), instead they talk about other things. Two examples are “number of samples measured” (Fig 1b,1c, 3d, S1g) and “total number of biological replicates for each genotype measured in independent experiments” (4c,e and 4h,i and S4e,f). In the former case, we are left wondering whether this is only one independent experiment -which would not generally be acceptable. In the latter case, we are wondering whether there are multiple n numbers from each independent experiment and several independent experiments all lumped together into one population. If this is the case, then good statistical practice would be to state clearly how many independent experiments were done and then to use an alternative statistical method that accounts for the different variances within and between experiments, such as mixed-effect models.

We have modified the presentation of all figures accordingly and indicated in each fig. legend the statistical test used.

We have repeated experiments of Fig. 1b,c and Fig.3d in order to compare independent experiments made on independent days. For All other figures, results are comparing different sets of experiments made independently and statistical tests used are mentioned.

We have now modified Suppl. Table 1 and specified when and how many independent experiments were used. For results emanating from several independent experiments, statistical significance was tested using ordinary two-way Anova with Sidak’s multiple comparisons test.